# Rankformer: a Ranking-Inspired Transformer Model for Recommendation

## Abstract

Recommender Systems (RS) aim to generate personalized ranked lists for each user and are also evaluated using ranking metrics. Although personalized ranking is a fundamental aspect of RS, this critical property is often overlooked in the design of model architectures. To address this issue, we propose Rankformer, a ranking-inspired recommendation model. The architecture of Rankformer is inspired by the gradient of the ranking objective, embodying a unique (graph) transformer architecture — it leverages global information from all users and items to produce more informative representations, and employs specific attention weights to guide the evolution of embeddings towards improved ranking performance. We further develop an acceleration algorithm for Rankformer, reducing its complexity to a linear level with respect to the number of positive instances. Extensive experimental results demonstrate that Rankformer outperforms state-of-the-art methods.

## CCS Concepts

• **Information systems → Recommender systems**.

## Keywords

Recommendation, Transformer, Personalized Ranking

**ACM Reference Format:**
Anonymous Author(s). 2018. Rankformer: a Ranking-Inspired Transformer Model for Recommendation . In *Proceedings of Make sure to enter the correct conference title from your rights confirmation emai (Conference acronym 'XX).* ACM, New York, NY, USA, 11 pages. https://doi.org/XXXXXXX.XXXXXXX

## 1 Introduction

Recommender systems (RS) have been integrated into many personalized services, playing an essential role in various applications [20]. A fundamental attribute that distinguishes RS from other machine learning tasks is its inherently *personalized ranking* nature. Specifically, RS aims to create user-specific ranked lists of items and retrieve the most relevant ones for recommendation [32]. To achieve this, most existing RS approaches adopt model-based paradigms, where a recommendation model is learned from users' historical interactions and subsequently generates ranking scores for each user-item pair.

Recent years have witnessed substantial progress in recommendation model architectures, evolving from basic matrix factorization

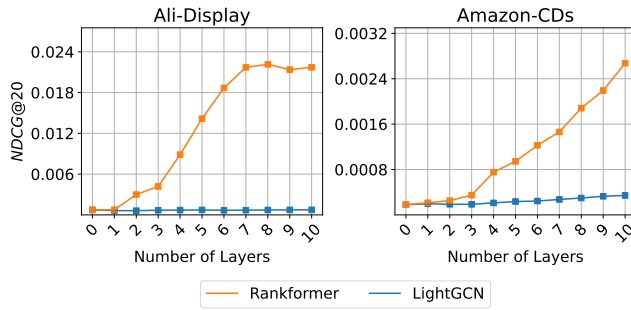

**Figure 1: Performance in terms of** $NDCG@20$ **when using Rankformer and LightGCN with different numbers of layers for randomly initialized representations without training.**

[21, 27] to more advanced architectures like Auto-Encoder [10, 25] and Graph Neural Networks (GNNs) [12, 39, 40, 49]. despite their increasingly sophisticated, we argue that a critical limitation remains — **their architecture design often neglects the essential ranking property of RS, which could compromise their effectiveness.**

To illustrate this point, let's take GNN-based architectures as examples for analyses. GNN-based models have been extensively studied in this field and usually achieve state-of-the-art performance [18]. In RS, the primary role of GNNs has been identified as a low-pass filter (*a.k.a.* graph smoother) [30], which draws the embeddings of connected users/items closer. However, this role significantly deviates from the ranking objective, thereby creating a gap that can impede their effectiveness. As demonstrated in Figure 1, without the guidance of a ranking objective function, the performance gains from stacking multiple layers of GNNs are limited. Even when supervised signals are introduced, stacking multiple GNN layers would easily result in the notorious issue of over-smoothing, where the score differences between positive and negative instances are reduced and become indistinguishable. This is in direct conflict with the ranking objective, which aims to elevate scores of the positive instances over the negative ones, leading to suboptimal performance. These limitations clearly underscore the necessity of considering the ranking property in the design of model architectures. It can serve as an *inductive bias* to guide the model towards generating better ranking performance. Naturally, a significant research question arises: **How can we develop a recommendation architecture that is well-aligned with the ranking property of RS?**

Towards this end, we propose a novel ranking-inspired recommendation model, **Rankformer**. Instead of a heuristic design or inheriting from other domains, the architecture of Rankformer is directly inspired by the ranking objective. Specifically, we scrutinize the gradient of the ranking objective, which suggests the

evolution direction of embeddings for enhanced ranking performance, and consequently develop a neural layer in accordance with this gradient. Rankformer embodies a unique (graph) transformer architecture: It leverages global information from all other users and items to obtain more informative user/item representations. Furthermore, specific attention weights, which compare positive and negative instances, are introduced to guide the evolution of embeddings towards improved ranking performance.

Despite its theoretical effectiveness, the implementation of Rankformer could encounter the challenge of computational inefficiency. As a transformer model, Rankformer involves information propagation between each user and item, leading to a time complexity that is quadratic in relation to the number of users and items. This renders it computationally prohibitive in practice. Thus, we propose an acceleration algorithm specifically tailored for Rankformer. By utilizing mathematical transformations and memorization skills, we reduce the complexity to linear with respect to the number of positive instances, significantly improving the model's efficiency.

Overall, this work makes the following contributions:

- Proposing a novel ranking-inspired recommendation model, Rankformer, that explicitly incorporates the personalized ranking principle into the model architecture design.
- Customizing a fast algorithm for Rankformer, reducing the original computational complexity from quadratic with the number of users and items to linear with the positive instances.
- Conducting experiments on four real-world datasets to demonstrate the superiority of Rankformer over state-of-the-art methods by a significant margin (4.74% on average).

## 2 Preliminary

In this section, we present the background of recommender systems and the transformer model.

### 2.1 Background on Recommender Systems

**Task Formulation.** This work focuses on collaborative filtering, a generic and common recommendation scenario. Given a recommender system with a set of users $\mathcal{U}$ and a set of items $\mathcal{I}$, let $n$ and $m$ denote the number of users and items. Let $\mathcal{D} = \{y_{ui} : u \in \mathcal{U}, i \in \mathcal{I}\}$ denote the historical interactions between users and items, where $y_{ui} = 1$ indicates that user $u$ has interacted with item $i$, and $y_{ui} = 0$ indicates has not. For convenience, we define $\mathcal{N}_u^+ = \{i \in \mathcal{I} : y_{ui} = 1\}$ as the set of positive items for user $u$, and $\mathcal{N}_u^- = \{\mathcal{I} \setminus \mathcal{N}_u^+\}$ as the negative item set. Similar definitions apply for $\mathcal{N}_i^+$ and $\mathcal{N}_i^-$ which indicate the set of positive and negative users for item $i$, respectively. The recommendation task can be formulated as learning a personalized ranked list of items for users and recommending the top items that users are most likely to interact with.

Personalized ranking is a fundamental property that distinguishes RS from other machine-learning tasks. RS is typically evaluated using ranking metrics such as NDCG@K, AUC, and Precision@K, which measure how well positive instances are ranked higher than negative ones [41]. Given this, the importance of considering the ranking property in model architecture design cannot be overstated.

**Recommendation Models.** Modern recommender systems typically model ranking scores using a learnable recommendation model. Embedding-based models are widely adopted [21, 32]. These models map the features (e.g., ID) of users and items into a $d$-dim embedding $\mathbf{z_u}, \mathbf{z_i} \in \mathbb{R}^d$, and generate their predicted score $\hat{y}_{ui}$ based on similarity of embeddings. The inner product, inherited from matrix factorization (MF), is commonly used in RS, i.e., $\hat{y}_{ui} = \mathbf{z}_u^T \mathbf{z}_i$, as it supports efficient retrieval and has demonstrated strong performance in various scenarios [2, 18]. The predicted scores $\hat{y}_{ui}$ are subsequently utilized to rank items for generating recommendations. For convenience, we collect the embeddings of all users and items as a matrix $\mathbf{Z}$.

Given the ranking nature of RS, existing recommendation models are often trained with ranking-oriented objective functions, e.g., BPR [32]. BPR aims to raise the scores of positive items relative to negative ones and can be formulated as:

$$\mathcal{L}_{BPR} = - \sum_{u \in \mathcal{U}} \sum_{i \in \mathcal{N}_u} \sum_{j \notin \mathcal{N}_u} \frac{\sigma(\mathbf{z}_u^T \mathbf{z}_i - \mathbf{z}_u^T \mathbf{z}_j)}{d_u(m - d_u)} + \lambda \|\mathbf{Z}\|_2^2 \quad (1)$$

where $\sigma(.)$ denotes the activation, and the hyperparameter $\lambda$ controls the strength of the regularizer. While BPR provides a supervised signal to guide model training toward better ranking, it does not negate the importance of model architecture. As demonstrated in recent machine learning literature [6], model architecture acts as an inductive bias that determines the model's learning and generalization capacity — if the bias aligns well with the underlying patterns of the task, the model is likely to perform well. In subsection , we conduct specific analyses to demonstrate the merits of incorporating ranking properties into model architecture design.

**GNN-based Recommendation Models.** In recent years, Graph Neural Networks (GNNs) have been widely explored in the field of RS and have demonstrated notable effectiveness [12, 17, 18, 40]. These methods construct a bipartite graph from historical interactions, where users and items are represented as nodes, and an edge exists between them if the user has interacted with the item. User/item embeddings are iteratively refined by aggregating information from their graph neighbors. Formally, taking the representative LightGCN [18] model as an example, the $l$-layer network can be written as:

$$\mathbf{z}_u^{(l)} = \sum_{i \in \mathcal{N}_u^+} \frac{1}{\sqrt{d_u d_i}} \mathbf{z}_i^{(l)}; \quad \mathbf{z}_i^{(l)} = \sum_{u \in \mathcal{N}_i^+} \frac{1}{\sqrt{d_u d_i}} \mathbf{z}_u^{(l)} \quad (2)$$

where $d_u = |\mathcal{N}_u^+|$, is the degree of node $u$ in graph, and $d_i = |\mathcal{N}_i^+|$.

Recent work has shown that GNNs act as low-pass filters, which is beneficial for capturing collaborative signals [53]. Nevertheless, as discussed earlier, there is a gap between the role of GNNs and the ranking objective, limiting their effectiveness. While the work [34] attempts to build theoretical connections, their finding is constrained by impractical assumptions, such as requiring large embedding spaces, single-layer GNNs, or untrained models.

### 2.2 Transformer Architecture

Transformer has been widely adopted in various fields [4, 35, 36]. The core of the Transformer is the attention module, which takes

input $X \in \mathbb{R}^{n \times d}$ and computes:

$$\mathbf{Q} = \mathbf{X}\mathbf{W}_Q, \quad \mathbf{K} = \mathbf{X}\mathbf{W}_K, \quad \mathbf{V} = \mathbf{X}\mathbf{W}_V,$$

$$\text{Attn}(\mathbf{X}) = \text{softmax}(\frac{\mathbf{Q}\mathbf{K}^T}{\sqrt{d_K}})\mathbf{V} \tag{3}$$

where $\mathbf{W}_Q \in \mathbb{R}^{d \times d_K}, \mathbf{W}_K \in \mathbb{R}^{d \times d_K}, \mathbf{W}_V \in \mathbb{R}^{d \times d_V}$ are weight matrices for the query, key, and value projections, respectively.

The transformer architecture has significant potential to be exploited in RS. Considering the input as embeddings of users and items, transformer estimates the similarity between these entities based on their projected embeddings, and then aggregates information from other entities according to this similarity. Such operations align closely with the fundamental principles of collaborative filtering [5, 23]. However, the architecture should be specifically designed to align with the ranking principle, and the high computational overhead from global aggregation needs to be addressed.

## 3 Methodology

In this section, we first introduce the architecture of the proposed Rankformer, followed by a discussion of its connections with existing architectures. Finally, we detail how its computation is accelerated.

### 3.1 Rankformer Layer

Rankformer is directly inspired by the ranking objective — we aim to promote user/item embeddings to evolve towards better ranking performance as they progress through the neural network. To achieve this, a natural idea is to let the embeddings evolve in alignment with the gradient of the ranking objective, which suggests the directions for enhancing ranking performance. Specifically, let $\mathcal{L}(\mathbf{Z})$ be a specific ranking objective. We may employ gradient descent (or ascent) to update the embeddings to improve their quality with:

$$\mathbf{Z}^{(l+1)} = \mathbf{Z}^{(l)} - \tau \cdot \frac{\partial \mathcal{L}(\mathbf{Z}^{(l)})}{\partial \mathbf{Z}^{(l)}} \tag{4}$$

where $\mathbf{Z}^{(l+1)}$ denotes the embeddings in the $l$-th step and $\tau$ denotes the step-size. This inspires us to develop a neural network that mirrors such an update. We may simply let $\mathbf{Z}^{(l+1)}$ be the embeddings of the $l$-layer and employ a neural network along with Eq(1). Naturally, the neural network would capture the ranking signals of the recommendation and guide the embeddings to evolve toward better ranking performance.

To instantiate this promising idea, this work simply adopts the conventional objective BPR for neural network design, as it has been demonstrated to be an effective surrogate for the AUC metric. Moreover, for facilitating analyses and accelerating computation, we simply choose the quadratic function activation, i.e., $\sigma(x) = x^2 + cx$. This can also be considered as a second-order Taylor approximation of other activations. The $l$-layer neural network of Rankformer can be formulated as follows by mirroring the gradient descent of BPR:

$$\mathbf{z}_u^{(l+1)} = (1-\tau)\mathbf{z}_u^{(l)} + \frac{\tau}{C_u^{(l)}}\left(\underbrace{\sum_{i \in \mathcal{N}_u^+} \Omega_{ui}^{+\,(l)}\mathbf{z}_i^{(l)}}_{\text{Aggregate Positive}} + \underbrace{\sum_{i \in \mathcal{N}_u^-} \Omega_{ui}^{-\,(l)}\mathbf{z}_i^{(l)}}_{\text{Aggregate Negative}}\right)$$

$$\mathbf{z}_i^{(l+1)} = (1-\tau)\mathbf{z}_i^{(l)} + \frac{\tau}{C_i^{(l)}}\left(\underbrace{\sum_{u \in \mathcal{N}_i^+} \Omega_{iu}^{+\,(l)}\mathbf{z}_u^{(l)}}_{\text{Aggregate Positive}} + \underbrace{\sum_{u \in \mathcal{N}_i^-} \Omega_{iu}^{-\,(l)}\mathbf{z}_u^{(l)}}_{\text{Aggregate Negative}}\right) \tag{5}$$

where $\Omega^+$ and $\Omega^-$ denote the weights for aggregating positive and negative users/items by items/users:

$$\Omega_{ui}^{+\,(l)} = \Omega_{iu}^{+\,(l)} = \frac{1}{d_u}\left(\underbrace{(\mathbf{z}_u^{(l)})^T\mathbf{z}_i^{(l)}}_{\text{Similarity}} - \underbrace{b_u^{-\,(l)}}_{\text{Benchmark}} + \underbrace{\alpha}_{\text{Offset}}\right)$$

$$\Omega_{ui}^{-\,(l)} = \Omega_{iu}^{-\,(l)} = \frac{1}{m-d_u}\left(\underbrace{(\mathbf{z}_u^{(l)})^T\mathbf{z}_i^{(l)}}_{\text{Similarity}} - \underbrace{b_u^{+\,(l)}}_{\text{Benchmark}} - \underbrace{\alpha}_{\text{Offset}}\right) \tag{6}$$

where $b_u^{+\,(l)}$ and $b_u^{-\,(l)}$ are two benchmark terms used to calculate the average similarity of positive/negative pairs.

$$b_u^{+\,(l)} = \frac{1}{d_u}\sum_{j \in \mathcal{N}_u^+}(\mathbf{z}_u^{(l)})^T\mathbf{z}_j^{(l)}$$

$$b_u^{-\,(l)} = \frac{1}{m-d_u}\sum_{j \in \mathcal{N}_u^-}(\mathbf{z}_u^{(l)})^T\mathbf{z}_j^{(l)} \tag{7}$$

Detailed derivations can be found in Appendix A.1. Here we simply set the hyperparameter $\lambda = 1$. Additionally, we introduce normalization constants to maintain numerical stability, i.e.,

$$C_u^{(l)} = \sum_{i \in \mathcal{N}_u^+}\left|\Omega_{ui}^{+\,(l)}\right| + \sum_{i \in \mathcal{N}_u^-}\left|\Omega_{ui}^{-\,(l)}\right|$$

$$C_i^{(l)} = \sum_{u \in \mathcal{N}_i^+}\left|\Omega_{iu}^{+\,(l)}\right| + \sum_{u \in \mathcal{N}_i^-}\left|\Omega_{iu}^{-\,(l)}\right| \tag{8}$$

While the neural layer may seem complex, its underlying intuition is straightforward. The derived Rankformer embodies a unique transformer architecture—each user (or item) iteratively leverages global information from all items (or users) to update its representation, utilizing both positive and negative interactions. For example, a positive item indicates what the user likes, while a negative item suggests what the user dislikes. Both signals are crucial for profiling user preferences. Aggregating information from both types of items helps guide the user's embedding closer to the items they like and away from the items they dislike. Similar logic applies to the item side, where aggregating information from both positive and negative users enhances the quality of item embeddings.

Furthermore, specific attention weights (i.e., , Eq(6)) are introduced, consisting of three parts:

- Similarity term $\mathbf{z}_u^T\mathbf{z}_i$: This operation can be seen as a vanilla attention mechanism akin to that in the Transformer model,

**Table 1: Comparison of Rankformer with LightGCN, GAT, and vanilla Transformer. These three methods can be viewed as special cases of Rankformer with certain components removed.**

| | $\Omega_{ui}^+$ | $\Omega_{ui}^-$ |
|---|---|---|
| LightGCN | $\frac{1}{d_u}$ | 0 |
| GAT | $\frac{z_u^T z_i}{d_u}$ | 0 |
| Transformer | $\frac{z_u^T z_i}{d_u}$ | $\frac{z_u^T z_i}{d_u}$ |
| Rankformer | $\frac{z_u^T z_i - b_u^- + \alpha}{d_u}$ | $\frac{z_u^T z_i - b_u^- - \alpha}{d_u}$ |

except that we omit the extra projection parameters. The intuition here is that similar entities provide more valuable information and thus should be given higher attention weights during information aggregation.

- Benchmark term $b_u^+$ or $b_u^-$: Rankformer incorporates additional benchmark terms that represent the average similarity among all positive (or negative) items for each user. This approach aligns with the inherent ranking nature of RS. The absolute similarity value (*i.e.,* the prediction score) may not always be the most important factor; rather, the relative score — *i.e.,* how much an item's score is higher (or lower) compared to others — provides crucial evidence of its ranking, indicating the degree of its positivity or negativity. Thus, for positive instances, a negative benchmark is used as a reference point for comparison, and a similar strategy is applied to negative instances.

- Offset term $\alpha$: This term differentiates the influence of positive and negative interactions. For positive user-item pairs, the weights are increased, bringing their embeddings closer, while for negative pairs, the weights are reduced (and can even become negative), pushing their embeddings apart. Additionally, the magnitude of $\alpha$ can act as a smoothing factor. A larger $\alpha$ makes the weight distribution among positive (or negative) instances more uniform while a smaller $\alpha$ sharpens the distribution.

## 3.2 Disscussion

In this subsection, we elucidate the distinctions and connections between Rankformer and existing methodologies. Table 1 summarizes these relationships, wherein LightGCN, GAT, and the vanilla Transformer can be viewed as special cases of Rankformer with certain components omitted.

**Comparison with GNNs-based methods:** When compared to existing GNNs-based methods (*e.g.,* LightGCN[18], GAT [37]), the primary distinction of Rankformer is its ability to leverage signals from negative instances during the aggregation process. This aspect is pivotal, as negative instances also provide valuable information for profiling user preferences or item attributes. For instance, negative items supply signals about user dislikes, which are equally valuable for learning user representations, *i.e.,* distancing the user's representation from the item.

**Comparison with Transformer:** When compared to existing (graph) transformer models [12, 18], Rankformer exhibits three differences: 1) During information aggregation, Rankformer handles positive and negative relations separately, calculating their attention weights in different manners. This treatment is rational, as they deliver distinctly different types of signals, enabling the transformer to perceive such crucial historical interaction information. 2) Beyond embedding similarity, Rankformer introduces additional benchmark and offset terms, ensuring the neural network aligns well with the ranking objective. 3) Most importantly, Rankformer is not heuristically designed but is entirely derived from the ranking objective, guiding the evolution of embeddings toward improved ranking performance.

**Comparison with Direct Optimization:** Note that a recommendation model is also optimized through a rank-oriented loss function, *e.g.,* BPR. Given this, some readers may question the advantages of incorporating gradient descent of ranking objectives into the design of the model architecture.

The architecture of Rankformer is designed to simulate multiple iterations of gradient descent from the original embeddings. This implies that Rankformer has already progressed along the optimization trajectory, thus gaining insight into the forthcoming optimization landscape and the potential of various optimization directions. Therefore, when Rankformer is trained with BPR, it naturally tends to select a more advantageous optimization direction than a conventional recommendation model. This is due to its advanced perspective of the optimization landscape and its inherent objective to enhance model performance after successive optimization steps.

## 3.3 Fast Implementation

As a transformer model, the implementation of Rankformer would also face an inefficiency challenge. The computational overhead primarily originates from its information aggregation mechanism, which involves global aggregation between users and items, with the complexity $O(nmd)$. Given the extensive number of users and items in real-world scenarios, such operations can be prohibitively expensive. The majority of the complexity originates from the aggregation of negative instances. However, these computationally demanding terms can be transformed with appropriate mathematical manipulations. Here, we take the term for aggregating $\sum_{j \in \mathcal{N}_u^-} z_j^{(l)}$ as an example:

$$\sum_{j \in \mathcal{N}_u^-} z_j^{(l)} = \sum_{j \in \mathcal{I}} z_j^{(l)} - \sum_{i \in \mathcal{N}_u^+} z_j^{(l)} \tag{9}$$

and the term $b_u^{-(l)}$ can be also fast calculated with:

$$b_u^{-(l)} = \frac{1}{m - d_u} (z_u^{(l)})^T \left( \sum_{j \in \mathcal{I}} z_j^{(l)} - \sum_{j \in \mathcal{N}_u^+} z_j^{(l)} \right) \tag{10}$$

The complexity of calculating these terms can be reduced to $O((n + m)d + Ed)$, where $E$ is the number of edges. Similar methods can be applied to calculating $\Omega_{ui}^{+(l)}$, $\Omega_{ui}^{-(l)}$, $C_u^{(l)}$, $C_i^{(l)}$ with complexity $O((n + m)d + Ed)$, and calculating $z_u^{(l+1)}$, $z_i^{(l+1)}$ with complexity $O((n + m)d^2 + Ed)$. Readers may refer to the Appendix A.2 for more detailed derivations and complexity analyses. With such an

Table 2: tab: Statistics of datasets.

| Dataset | #User | #Item | #Interaction |
|---|---|---|---|
| Ali-Display | 17,730 | 10,036 | 173,111 |
| Amazon-Kindle | 47,754 | 47,052 | 689,550 |
| Amazon-CDs | 51,266 | 46,463 | 731,734 |
| Yelp2018 | 167,037 | 79,471 | 1,970,721 |

algorithm, although Rankformer involves propagation between every user-item pair, the complexity of Rankformer can be reduced to $O((n + m)d^2 + Ed)$, making it highly efficient and applicable.

We also introduce a few simple strategies during implementation to further enhance training stability: 1) During the calculation of attention weights, we employ embedding normalization. This procedure constrains the value of similarity within a fixed range of [-1,1]. 2) Given that initial embeddings may not be of high quality and could potentially affect the calculation of attention, we employ a warm-up strategy. Specifically, we employ uniform weights and remove negative aggregation at the first layer of Rankformer.

## 4 Experiments

In this section, we conduct comprehensive experiments to answer the following research questions:

- **RQ1:** How does Rankformer perform compared to existing state-of-the-art methods?
- **RQ2:** What are the impacts of the important components (such as the terms in Eq(6) on Rankformer?
- **RQ3:** How do the hyperparameters affect the performance of Rankformer?
- **RQ4:** How does the efficiency of Rankformer compare with existing methods?

### 4.1 Experimental Settings

*4.1.1 Datasets.* We conducted experiments on four conventional real-world datasets: **Ali-Display** [1] provided by Alibaba, is a dataset for estimating click-through rates of Taobao display ads; **Amazon-Kindle** and **Amazon-CDs** [28] consist of user ratings on products on the Amazon platform; **Yelp2018** [2] is a dataset of user reviews collected by Yelp. We adopt a standard 5-core setting and randomly split the datasets into training, validation, and test sets in a 7:1:2 ratio. The statistical information of the datasets is presented in Table 2.

*4.1.2 Metrics.* We closely refer to [18, 50] and employed two widely used metrics, *Recall@K* and *NDCG@K*, to evaluate the recommendation accuracy. We also simply set $K = 20$ as recent work [50].

*4.1.3 Baselines.* **1) Recommendation Methods.** The following five representative or SOTA recommendation methods are included:
- **MF** [21]: the method exclusively employs the BPR loss function for matrix factorization, without any encoding architecture.
- **LightGCN** [18]: the classical recommendation method that adopts linear GNN.

[1]https://tianchi.aliyun.com/dataset/dataDetail?dataId=56
[2]https://www.yelp.com/dataset

- **DualVAE** [16]: a collaborative recommendation method that combines disentangled representation learning with variational inference.
- **MGFormer** [5]: the state-of-the-art recommendation method based on the graph transformer, equipped with a masking mechanism designed for large-scale graphs.
- **CAGCN** [40]: a recommendation-tailored GNN, introducing the recommendation-oriented topological metric CIR.

**2) Graph Transformer with Global Attention Mechanism.** The following three state-of-the-art graph transformer methods are included. We augment these methods with BPR loss to apply them in recommendation tasks:
- **Nodeformer** [44]: the classical linear transformer for large-scale graphs with a kernelized Gumbel-Softmax operator to reduce the algorithmic complexity.
- **DIFFormer** [14]: the linear transformer derived from an energy-constrained diffusion model, applicable to large-scale graphs. This work is built upon NodeFormer.
- **SGFormer** [45]: the latest linear Transformer designed for large-scale graphs, built upon NodeFormer and DIFFormer.

**3) Recommendation Methods With Contrastive Learning Loss.** Given the SOTA methods are often achieved with contrastive learning, we also compare Rankformer with these methods. Nevertheless, it would be unfair as our Rankformer does not utilize contrastive loss. Thus, we also equipped Rankformer with layer-wise constrastive loss as XSimGCL [50] (named as Rankformer-CL). The following baselines are adopted:
- **XSimGCL** [50]: the state-of-the-art method that enhance Light-GCN with contrastive learning.
- **GFormer** [23]: the state-of-the-art recommendation method that combine transformer architecture with contrastive learning.

*4.1.4 Parameter Settings.* For Rankformer, we adopt the Adam optimizer and search the hyperparameters with grid search. Specifically, we set the hidden embedding dimension $d = 64$ as recent works [18]. The weight decay is set to $1e - 4$. We search for $\tau$ with a step size of 0.1 within the range $[0, 1]$, and select the number of layers $L$ for Rankformer in the range of $\{1, 2, 3, 4, 5\}$. To reduce the number of hyperparameters, except for the experiments in Table 3, the parameter $\alpha$ is simply set to 2, although Table 3 indicates that fine-tuning this parameter could improve the model's performance. For Rankformer without contrastive learning loss, the learning rate is set to 0.1.

We also test Rankformer-CL that combines Rankformer with the layer-wise contrastive loss used in XSimGCL. For Rankformer-CL, the batch size is set to 2048, and the learning rate is set to 0.001. The relevant parameters for the contrastive loss are simply set as: $\epsilon_{\text{cl}} = 0.2$, $\lambda_{\text{cl}} = 0.05$, $\tau_{\text{cl}} = 0.15$.

For the compared methods, we use the source code provided officially and follow the instructions from the original paper to search for the optimal hyperparameters. We extensively traversed and expanded the entire hyperparameter space as recommended by the authors to ensure that all compared methods achieved optimal performance.

**Table 3: Performance comparison between baselines and Rankformer. The best result is bolded and the runner-up is underlined. The mark '*' suggests the improvement is statistically significant with $p < 0.05$.**

|  |  | Ali-Display | | Amazon-Kindle | | Amazon-CDs | | Yelp2018 | |
|---|---|---|---|---|---|---|---|---|---|
|  |  | ndcg@20 | recall@20 | ndcg@20 | recall@20 | ndcg@20 | recall@20 | ndcg@20 | recall@20 |
| Recommendation Methods | MF | 0.0586 | 0.1094 | 0.1092 | 0.1764 | 0.0717 | 0.1255 | 0.0402 | 0.0766 |
|  | LightGCN | 0.0643 | 0.1174 | 0.1273 | 0.2016 | 0.0764 | 0.1317 | 0.0456 | 0.0842 |
|  | DualVAE | 0.0603 | 0.1119 | 0.1111 | 0.1786 | 0.0733 | 0.1276 | 0.0402 | 0.0768 |
|  | CAGCN | 0.0645 | 0.1174 | 0.1320 | 0.2049 | 0.0797 | 0.1322 | 0.0431 | 0.0787 |
|  | MGFormer | 0.0649 | 0.1083 | 0.1249 | 0.1980 | 0.0763 | 0.1324 | 0.0468 | 0.0869 |
| Graph Transformer | Nodeformer | 0.0205 | 0.0358 | - | - | - | - | - | - |
|  | DIFFormer | 0.0370 | 0.0715 | 0.0448 | 0.0804 | 0.0369 | 0.0681 | 0.0282 | 0.0539 |
|  | SGFormer | 0.0411 | 0.0769 | 0.0277 | 0.0518 | 0.0284 | 0.0524 | 0.0218 | 0.0427 |
| Rankformer | | **0.0652*** | **0.1208*** | **0.1379*** | **0.2092*** | **0.0831*** | **0.1412*** | **0.0482*** | **0.0890*** |
|  | | 0.42% | 2.87% | 4.48% | 2.09% | 4.27% | 6.64% | 2.90% | 2.46% |
| Graph-based RS with CL | XSimGCL | 0.0650 | 0.1194 | 0.1287 | 0.1999 | 0.0796 | 0.1346 | 0.0500 | 0.0907 |
|  | Gformer | 0.0644 | 0.1176 | 0.1303 | 0.1981 | 0.0812 | 0.1366 | 0.0502 | 0.0897 |
| Rankformer-CL | | **0.0680*** | **0.1246*** | **0.1445*** | **0.2159*** | **0.0877*** | **0.1467*** | **0.0523*** | **0.0941*** |
|  | | 4.57% | 4.32% | 10.93% | 8.00% | 8.00% | 7.41% | 4.29% | 3.71% |

## 4.2 Performance Comparison (RQ1)

The performance comparison of our Rankformer and all baselines in terms of *Recall*@20 and *NDCG*@20 is shown in Table 3. Overall, Rankformer consistently outperforms all comparison methods across all datasets with an average of improvements of 4.74%. This result clearly demonstrates the effectiveness of leveraging ranking signals in model architecture.

**Comparison with Recommendation Methods.** Overall, our Rankformer outperforms existing state-of-the-art recommendation methods. Among these comparative methods, graph-based recommender methods such as LightGCN, CAGCN, and MGFormer outperform other methods, indicating the advantage of using graph structure in the recommendation. GFormer, which incorporates transformer architecture, outperforms LightGCN and CAGCN using traditional graph GCNs, suggesting that transformer architectures can better leverage collaborative information.

**Comparison with Graph Transformer Methods.** Rankformer consistently outperforms all transformer-based graph representation methods across all datasets. Remarkably, these baseline performances often fall below standard benchmarks and even fail to converge on some datasets, indicating their unsuitability for recommendation tasks. There are two key factors contributing to this outcome: 1) These methods are designed for traditional graph tasks such as node classification, hence they are also based on the smoothness assumption, which does not align with the ranking objectives of recommendation. 2) These methods typically employ a large number of parameters and non-linear modules, making it challenging to effectively train them in RS due to data sparsity.

**Comparison with Graph-based RS with contrastive learning.** The performance of the methods with contrastive learning surpasses other baselines significantly, and adding the contrastive learning loss to Rankformer further enhances recommendation performance noticeably. The Rankformer augmented with the CL-loss notably outperforms these CL-based methods. Particularly on the "Amazon-Kindle" dataset, the improvement reaches an impressive

10.93%. These observations indicate that contrastive learning can effectively leverage rich collaborative information, and our Rankformer model integrates well with contrastive learning modules.

## 4.3 Ablation Study (RQ2)

To investigate the effects of each module in Rankformer, we conduct an ablation study and the results are presented in Table 4. We draw the following observations:

By removing information aggregation between negative pairs, we observe significantly performance drops. This is coincident with our intuition, as the negative relations also bring valuable signals to learn user or item representations;

By removing benchmark terms $b_u$, we also observe performance drops, with an exception on the dataset "Amazon-Kindle". This can be partly attributed to our simple fixed of $\alpha$, rather than fine-tuning to find its optimal. Figure 3 indicates that the optimal $\alpha$ is approximately 6 rather 2. Therefore, on "Amazon-Kindle", the removal of the benchmark module would relatively amplify the influence of $\alpha$, resulting in performance boost. Conversely, on larger and sparser datasets like "Yelp2018", the role of the benchmark module becomes more pronounced.

By removing the normalization terms $C$ in Rankformer, we observe terrible performance of Rankformer. It would be trained unstable and usually suffers from gradient explosion. Thus, we introduce normalization terms in the Rankformer, as other transformer models do.

## 4.4 Role of the parameters (RQ3)

**Hyperparameter $\tau$ and the number of Rankformer layers $L$.** The parameter $\tau$ is proportional to the step size used in simulating gradient ascent within Rankformer layers. As illustrated in Figure 2, the performance of Rankformer varies with $\tau$ as the number of layers changes, showing an initial increase followed by a decrease. This trend arises because, with a fixed number of layers, an excessively small step size can result in incomplete optimization, while a

**Table 4: The results of the ablation study, where negative pairs, benchmark terms and normalization terms have been removed, respectively.**

| | Ali-Display | | Amazon-Kindle | | Amazon-Cds | | Yelp2018 | |
|---|---|---|---|---|---|---|---|---|
| | ndcg@20 | recall@20 | ndcg@20 | recall@20 | ndcg@20 | recall@20 | ndcg@20 | recall@20 |
| Rankformer - w/o negative pairs | 0.0535 | 0.1026 | 0.1315 | 0.2013 | 0.0744 | 0.1295 | 0.0429 | 0.0800 |
| Rankformer - w/o benchmark | 0.0641 | 0.1189 | 0.1379 | **0.2094** | 0.0827 | 0.1406 | 0.0475 | 0.0884 |
| Rankformer - w/o normalization of $\Omega$ | 0.0291 | 0.0527 | 0.0610 | 0.1133 | 0.0357 | 0.0596 | 0.0166 | 0.0321 |
| Rankformer | **0.0652** | **0.1208** | **0.1379** | 0.2092 | **0.0831** | **0.1412** | **0.0482** | **0.0890** |

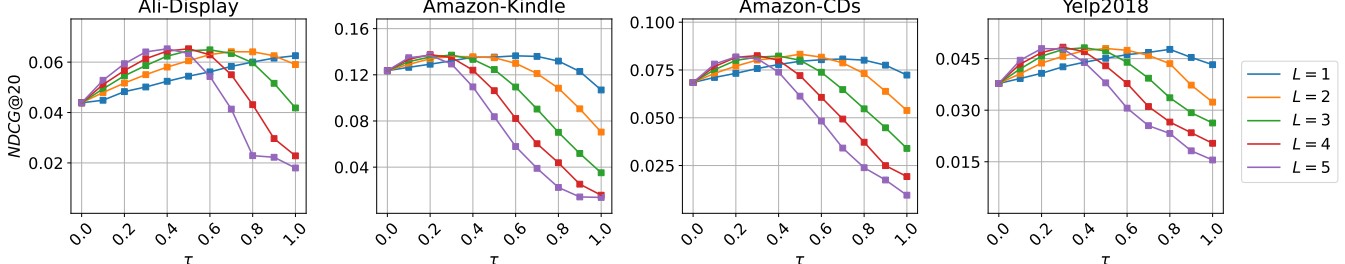

**Figure 2: Performance of Rankformer in terms of $NDCG@20$ with different layers $L$ and hyperparameter $\tau$.**

step size that is too large may lead to missing the optimal solution. Furthermore, as the number of Rankformer layers $L$ increases, the optimal value of $\tau$ decreases. This is because each Rankformer layer corresponds to one step of gradient ascent. Therefore, with a higher number of steps, approaching the optimal solution gradually with smaller steps is more effective; whereas with fewer steps, a larger step size is required to converge faster towards the optimal solution.

**Hyperparameter $\alpha$.** The parameter $\alpha$ controls the coefficient of the linear term in the Taylor expansion. Different values of $\alpha$ correspond to different activation functions $\delta(\cdot)$. As shown in Figure 3 , when $\alpha$ increases, the performance of Rankformer generally shows an initial improvement followed by a decline. This is because a larger $\alpha$ can better distinguish between positive and negative pairs, but an excessively large $\alpha$ can lead to over-smoothing of the representations learned by Rankformer. When $\alpha$ approaches positive infinity, the model will degenerate into a GCN that aggregates only the average representation of the entire graph and the representation of node neighborhoods.

### 4.5 Efficiency Comparison (RQ5)

Table 5 illustrates the actual runtime comparison between Rankformer and other baselines on four datasets, along with the theoretical computation time of Transformer with full-graph attention mechanism calculated on GPU using matrix blocking. The official implementation of LightGCN exhibits low efficiency, prompting us to introduce a re-implemented version of LightGCN for comparison. This version aligns with our Rankformer source code in data processing, training, testing, and other aspects, differing only in the encoding architecture. The experiments demonstrate that our Rankformer, with a complexity of $O((n + m)d^2 + Ed)$, exhibits similar actual runtime to LightGCN, which has a complexity of $O(Ed)$. In comparison to recommendation methods with graph transformer

such as MGFormer and GFormer, Rankformer exhibits significantly faster runtime efficiency.

## 5 Related Work

### 5.1 Architectures of Recommendation Models

In recent years, there has been a surge of publications on recommendation model architectures, ranging from traditional matrix factorization [19, 21, 29], neighbor-based methods [38], to more advanced auto-encoders [10, 25, 48], diffusion models [43], recurrent neural networks [9, 15], graph neural networks [18], and Transformer-based methods. In this section, we focus primarily on reviewing the most relevant GNN-based and Transformer-based recommendation models and refer readers to excellent surveys for more comprehensive information [1, 13, 33, 52].

Graph Neural Networks (GNNs), which leverage message-passing mechanisms to fully exploit collaborative information within graphs, have demonstrated remarkable effectiveness in the recommendation systems (RS) domain in recent years. Early work [12, 17, 39, 49] directly inherits the GNN architecture, including complex parameters, from the graph learning domain. Later, the representative LightGCN [18] pruned unnecessary parameters from GNNs, yielding better and more efficient performance. Building on LightGCN, recent work has made various improvements. For instance, some researchers [2, 42, 50, 51] have explored the use of contrastive learning strategies in GNN-based methods, achieving state-of-the-art performance; others [40] have refined and reweighted graph structures to better suit the recommendation task. Additionally, some works [31] have analyzed and enhanced LightGCN from a spectral perspective. While these GNN-based methods have achieved significant success, a potential limitation is that the role of GNNs

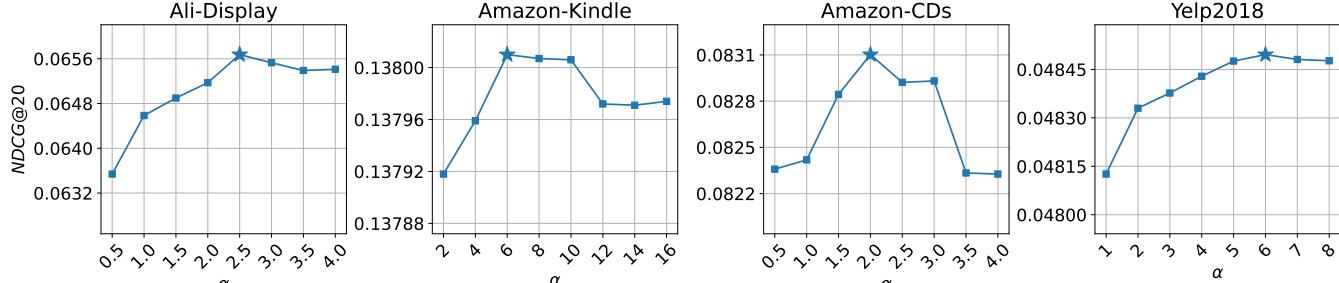

Figure 3: Performance in terms of $NDCG@20$ with different hyperparameter $\alpha$.

Table 5: Runtime comparison of Rankformer with baselines (in seconds).

|  | Ali-Display | Amazon-Kindle | Amazon-Cds | Yelp2018 |
|---|---|---|---|---|
| LightGCN - source code | 1687 | 8848 | 11694 | 21651 |
| LightGCN - rewrite | 99 | 132 | 91 | 1132 |
| MGFormer | 1192 | 2695 | 3090 | 2435 |
| Gformer | 2250 | 8987 | 6779 | 36587 |
| Rankformer | 109 | 246 | 282 | 2212 |

tends to deviate from the ranking objective, which may hinder their effectiveness.

Regarding the Transformer architecture, it was initially introduced to sequential recommendation as a superior alternative to recurrent neural networks [35] for modeling item dependencies in sequences. Additionally, Transformers have been employed in multimodal recommendation tasks to better fuse multimodal information [26]. However, these Transformer architectures differ significantly from our proposed Rankformer, as they are not tailored for generic recommendation scenarios and do not leverage global aggregation across all users and items. To the best of our knowledge, there are three related works that explore generic recommendation: GFormer [23], SIGformer [8], and MGFormer [5]. *We argue that the major limitations of these methods are heuristic designs that do not incorporate the ranking property into the architecture.* Moreover, they exhibit additional limitations: 1) SIGformer requires signed interaction information, which may not always be available; 2) Gformer utilizes the Transformer for generating contrastive views, rather than as the recommendation backbone; 3) MGFormer involves complex masking operations and positional encoding, making it difficult to train effectively and less efficient.

## 5.2 Graph Transformer

In recent years, there has been an increasing number of works applying Transformer to graph learning. By utilizing its global attention mechanism, Transformer enables each node on the graph to aggregate information from all nodes in the graph at each step, mitigating issues such as over-smoothing, over-squeezing, and expressive boundary problems caused by traditional GNNs that only aggregate low-order neighbors [46]. Research on graph Transformers mainly focused on designing positional encodings for capturing

topological structures, including spectral encoding [11, 22], centrality encoding [47], shortest path encoding [24, 47], and substructure encoding [3], which are suitable for positional encoding in graphs.

Given the time and space complexities of Transformers with global pairwise attention are usually proportional to the square of the number of nodes, the early study on graph transformer can only limited to small graphs. To tackle this, various acceleration strategies have been developed. NodeFormer [44] replaces the original attention matrix computation with a positive definite kernel to achieve linear computational complexity. Methods like Gophormer [55], ANS-GT [54], and NAGphormer [7] reduce computational complexity through sampling strategies, while DIFFormer [43] derives a linearizable Transformer using a diffusion model.

## 6 Conclusions

Personalized ranking is a fundamental attribute of Recommender Stystems (RS). In this work, we propose Rankformer, which explicitly incorporates this crucial property into its architectural design. Rankformer simulates the gradient descent process of the ranking objective and introduces a unique Transformer architecture. This specific design facilitates the evolution of embeddings in a direction that enhances ranking performance. In the future, it would be of great interest to extend this architecture to other recommendation scenarios, such as sequential recommendation and LLM-based recommendation, allowing ranking signals to be seamlessly integrated into the architectures.

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

# A  Appendices

## A.1  Derivation of Rankformer Layer

According to Eq(1), we have:

$$\mathcal{L}(\mathbf{Z};\sigma) = -\sum_{u \in \mathcal{U}} \sum_{i \in \mathcal{N}_u} \sum_{j \notin \mathcal{N}_u} \frac{\sigma(\mathbf{z}_u^T \mathbf{z}_i - \mathbf{z}_u^T \mathbf{z}_j)}{d_u(m-d_u)} + \lambda \|\mathbf{Z}\|_2^2 \quad (11)$$

By approximating the function $\sigma(x)$ using a second-order Taylor expansion, we obtain:

$$\widetilde{\mathcal{L}}(\mathbf{Z};\sigma) = -\sum_{u \in \mathcal{U}} \sum_{i \in \mathcal{N}_u} \sum_{j \notin \mathcal{N}_u} \frac{1}{d_u(m-d_u)} [\omega_{uij}(\mathbf{z}_u^T \mathbf{z}_i + \mathbf{z}_u^T \mathbf{z}_j)]$$
$$+ \lambda \|\mathbf{Z}\|_2^2 \quad (12)$$

where $\omega_{uij} = \mathbf{z}_u^T \mathbf{z}_i - \mathbf{z}_u^T \mathbf{z}_j + \alpha$. $\alpha > 0$ is a hyperparameter that controls the coefficient of the linear term in the Taylor expansion of $\sigma(.)$.

The optimization objective is to minimize the function $\mathcal{L}(\mathbf{Z};\sigma)$. Therefore, we propose performing one step gradient descent with the step size of $\tau$ on $\widetilde{\mathcal{L}}(\mathbf{Z};\sigma)$ within each Rankformer layer:

$$\mathbf{Z}^{(l+1)} = \mathbf{Z}^{(l)} + \tau \cdot \frac{\partial}{\partial \mathbf{Z}^{(l)}} \widetilde{\mathcal{L}}(\mathbf{Z}^{(l)};\sigma) \quad (13)$$

$$= \mathbf{Z}^{(l)} + \tau \cdot (\Omega^{(l)}\mathbf{Z} - \lambda \mathbf{Z}^{(l)}) \quad (14)$$

$$= (1 - \tau\lambda)\mathbf{Z}^{(l)} + \tau \cdot \Omega^{(l)}\mathbf{Z}^{(l)} \quad (15)$$

where $\Omega_{ui} = \Omega_{iu} = \begin{cases} \sum_{j \notin \mathcal{N}_u} \frac{\omega_{uij}}{d_u(m-d_u)}, & i \in \mathcal{N}_u \\ -\sum_{j \in \mathcal{N}_u} \frac{\omega_{uji}}{d_u(m-d_u)}, & i \notin \mathcal{N}_u \end{cases}$.

We simply set the hyperparameter $\lambda = 1$, so we get

$$\mathbf{Z}^{(l+1)} = (1 - \tau)\mathbf{Z}^{(l)} + \tau\Omega^{(l)}\mathbf{Z}^{(l)} \quad (16)$$

By reorganization the terms, we can easily get the Eq.(5).

## A.2  Time Complexity

Eq(5) - Eq(6) are computed as follows, achieving a complexity of $O((n+m)d^2 + Ed)$.

**Step 1:** Calculate $b_u^{+(l)}$ and $b_u^{-(l)}$ with a complexity of $O(n + md + Ed)$.

$$b_u^{+(l)} = \frac{1}{d_u} \sum_{j \in \mathcal{N}_u^+} (\mathbf{z}_u^{(l)})^T \mathbf{z}_j^{(l)} \quad (17)$$

$$= \frac{1}{d_u} (\mathbf{z}_u^{(l)})^T \left( \sum_{j \in \mathcal{N}_u^+} \mathbf{z}_j^{(l)} \right) \quad (18)$$

$$b_u^{-(l)} = \frac{1}{m - d_u} \sum_{j \in \mathcal{N}_u^-} (\mathbf{z}_u^{(l)})^T \mathbf{z}_j^{(l)} \quad (19)$$

$$= \frac{1}{m - d_u} (\mathbf{z}_u^{(l)})^T \left( \sum_{j \in \mathcal{N}_u^-} \mathbf{z}_j^{(l)} \right) \quad (20)$$

$$= \frac{1}{m - d_u} (\mathbf{z}_u^{(l)})^T \left( \sum_{j \in \mathcal{I}} \mathbf{z}_j^{(l)} - \sum_{j \in \mathcal{N}_u^+} \mathbf{z}_j^{(l)} \right) \quad (21)$$

Similarly, in the subsequent derivation, $\sum_{i \in \mathcal{N}_u^-} \mathbf{x}_i$ can be transformed into $\sum_{i \in \mathcal{I}} \mathbf{x}_i - \sum_{i \in \mathcal{N}_u^+} \mathbf{x}_i$ and computed linearly. The term $\sum_{u \in \mathcal{N}_i^-} \mathbf{x}_u$ can be transformed into $\sum_{u \in \mathcal{U}} \mathbf{x}_u - \sum_{u \in \mathcal{N}_i^+} \mathbf{x}_u$. Subsequently, the terms $\sum_{i \in \mathcal{N}_u^-}$ and $\sum_{u \in \mathcal{N}_i^-}$ will be retained without expansion.

**Step 2:** Calculate $C_u^{(l)}, C_i^{(l)}$, with a complexity of $O(n + md + Ed)$.

Since we have normalized $\mathbf{z}$, $(\mathbf{z}_u^{(l)})^T \mathbf{z}_i^{(l)} \leq 1$. When $\alpha \geq 2$, $\Omega_{ui}^{+(l)} = \Omega_{iu}^{+(l)} > 0$, and $\Omega_{ui}^{+(l)} = \Omega_{iu}^{+(l)} < 0$, so we can remove the absolute value symbols.

$$\sum_{i \in \mathcal{N}_u^+} \left| \Omega_{ui}^{+(l)} \right| = \sum_{i \in \mathcal{N}_u^+} \frac{(\mathbf{z}_u^{(l)})^T \mathbf{z}_i^{(l)} - b_u^{-(l)} + \alpha}{d_u} \quad (22)$$

$$= \frac{(\mathbf{z}_u^{(l)})^T}{d_u} \left( \sum_{i \in \mathcal{N}_u^+} \mathbf{z}_i^{(l)} \right) - b_u^{-(l)} + \alpha \quad (23)$$

$$\sum_{i \in \mathcal{N}_u^-} \left| \Omega_{ui}^{-(l)} \right| = -\sum_{i \in \mathcal{N}_u^-} \frac{(\mathbf{z}_u^{(l)})^T \mathbf{z}_i^{(l)} - b_u^{+(l)} - \alpha}{m - d_u} \quad (24)$$

$$= -\frac{(\mathbf{z}_u^{(l)})^T}{m - d_u} \left( \sum_{i \in \mathcal{N}_u^-} \mathbf{z}_i^{(l)} \right) + b_u^{+(l)} + \alpha \quad (25)$$

$$= 2\alpha - C_u^{+(l)} \quad (26)$$

$$\sum_{u \in \mathcal{N}_i^+} \left| \Omega_{iu}^{+(l)} \right| = \sum_{u \in \mathcal{N}_i^+} \frac{(\mathbf{z}_u^{(l)})^T \mathbf{z}_i^{(l)} - b_u^{-(l)} + \alpha}{d_u} \quad (27)$$

$$= (\mathbf{z}_i^{(l)})^T \left( \sum_{u \in \mathcal{N}_i^+} \frac{\mathbf{z}_u^{(l)}}{d_u} \right) - \left( \sum_{u \in \mathcal{N}_i^+} \frac{b_u^{-(l)}}{d_u} \right)$$
$$+ \alpha \left( \sum_{u \in \mathcal{N}_i^+} \frac{1}{d_u} \right) \quad (28)$$

$$\sum_{u \in \mathcal{N}_i^-} \left| \Omega_{iu}^{-(l)} \right| = -\sum_{u \in \mathcal{N}_i^-} \frac{(\mathbf{z}_u^{(l)})^T \mathbf{z}_i^{(l)} - b_u^{+(l)} - \alpha}{m - d_u} \quad (29)$$

$$= -(\mathbf{z}_i^{(l)})^T \left( \sum_{u \in \mathcal{N}_i^-} \frac{\mathbf{z}_u^{(l)}}{m - d_u} \right) + \left( \sum_{u \in \mathcal{N}_i^-} \frac{b_u^{+(l)}}{m - d_u} \right)$$
$$+ \alpha \left( \sum_{u \in \mathcal{N}_i^-} \frac{1}{m - d_u} \right) \quad (30)$$

**Step 3:** Calculate $\sum\limits_{i \in \mathcal{N}_u^+} \Omega_{ui}^{+\,(l)} \mathbf{z}_i^{(l)}$, $\sum\limits_{i \in \mathcal{N}_u^-} \Omega_{ui}^{-\,(l)} \mathbf{z}_i^{(l)}$, $\sum\limits_{u \in \mathcal{N}_i^+} \Omega_{iu}^{+\,(l)} \mathbf{z}_u^{(l)}$

and $\sum\limits_{u \in \mathcal{N}_i^-} \Omega_{iu}^{-\,(l)} \mathbf{z}_u^{(l)}$ with a complexity of $O(n + d^2 + Ed)$.

$$\sum_{i \in \mathcal{N}_u^+} \Omega_{ui}^{+\,(l)} \mathbf{z}_i^{(l)} = \sum_{i \in \mathcal{N}_u^+} \frac{(\mathbf{z}_u^{(l)})^T \mathbf{z}_i^{(l)} - b_u^{-\,(l)} + \alpha}{d_u} \mathbf{z}_i^{(l)} \quad (31)$$

$$= \frac{1}{d_u} \left( \sum_{i \in \mathcal{N}_u^+} (\mathbf{z}_u^{(l)})^T \mathbf{z}_i^{(l)} \mathbf{z}_i^{(l)} \right)$$

$$- \frac{b_u^{-\,(l)} - \alpha}{d_u} \left( \sum_{i \in \mathcal{N}_u^+} \mathbf{z}_i^{(l)} \right) \quad (32)$$

$$\sum_{i \in \mathcal{N}_u^-} \Omega_{ui}^{-\,(l)} \mathbf{z}_i^{(l)} = \sum_{i \in \mathcal{N}_u^-} \frac{(\mathbf{z}_u^{(l)})^T \mathbf{z}_i^{(l)} - b_u^{+\,(l)} - \alpha}{m - d_u} \mathbf{z}_i^{(l)} \quad (33)$$

$$= \frac{1}{m - d_u} \left( \sum_{i \in \mathcal{N}_u^-} (\mathbf{z}_u^{(l)})^T \mathbf{z}_i^{(l)} \mathbf{z}_i^{(l)} \right)$$

$$- \frac{b_u^{+\,(l)} + \alpha}{m - d_u} \left( \sum_{i \in \mathcal{N}_u^-} \mathbf{z}_i^{(l)} \right) \quad (34)$$

$$\sum_{u \in \mathcal{N}_i^+} \Omega_{iu}^{+\,(l)} \mathbf{z}_u^{(l)} = \sum_{u \in \mathcal{N}_i^+} \frac{(\mathbf{z}_u^{(l)})^T \mathbf{z}_i^{(l)} - b_u^{-\,(l)} + \alpha}{d_u} \mathbf{z}_u^{(l)} \quad (35)$$

$$= \left( \sum_{u \in \mathcal{N}_i^+} \frac{(\mathbf{z}_u^{(l)})^T \mathbf{z}_i^{(l)} \mathbf{z}_u^{(l)}}{d_u} \right)$$

$$- \left( \sum_{u \in \mathcal{N}_i^+} \frac{(b_u^{-\,(l)} - \alpha) \mathbf{z}_u^{(l)}}{d_u} \right) \quad (36)$$

$$\sum_{u \in \mathcal{N}_i^-} \Omega_{iu}^{-\,(l)} \mathbf{z}_u^{(l)} = \sum_{u \in \mathcal{N}_i^-} \frac{(\mathbf{z}_u^{(l)})^T \mathbf{z}_i^{(l)} - b_u^{+\,(l)} - \alpha}{m - d_u} \mathbf{z}_u^{(l)} \quad (37)$$

$$= \left( \sum_{u \in \mathcal{N}_i^-} \frac{(\mathbf{z}_u^{(l)})^T \mathbf{z}_i^{(l)} \mathbf{z}_u^{(l)}}{m - d_u} \right)$$

$$- \left( \sum_{u \in \mathcal{N}_i^-} \frac{(b_u^{+\,(l)} + \alpha) \mathbf{z}_u^{(l)}}{m - d_u} \right) \quad (38)$$

where $\sum\limits_{i \in \mathcal{N}_u^-} (\mathbf{z}_u^{(l)})^T \mathbf{z}_i^{(l)} \mathbf{z}_i^{(l)}$ and $\sum\limits_{u \in \mathcal{N}_i^-} \frac{(\mathbf{z}_u^{(l)})^T \mathbf{z}_i^{(l)} \mathbf{z}_u^{(l)}}{m - d_u}$ can be calculated as follows with a complexity of $O((n + m)d^2 + Ed)$:

$$\sum_{i \in \mathcal{N}_u^-} (\mathbf{z}_u^{(l)})^T \mathbf{z}_i^{(l)} \mathbf{z}_i^{(l)}$$

$$= \left( \sum_{i \in \mathcal{I}} (\mathbf{z}_u^{(l)})^T \mathbf{z}_i^{(l)} \mathbf{z}_i^{(l)} \right) - \left( \sum_{i \in \mathcal{N}_u^+} (\mathbf{z}_u^{(l)})^T \mathbf{z}_i^{(l)} \mathbf{z}_i^{(l)} \right) \quad (39)$$

$$= (\mathbf{z}_u^{(l)})^T \left( \sum_{i \in \mathcal{I}} \mathbf{z}_i^{(l)} \mathbf{z}_i^{(l)} \right) - \left( \sum_{i \in \mathcal{N}_u^+} (\mathbf{z}_u^{(l)})^T \mathbf{z}_i^{(l)} \mathbf{z}_i^{(l)} \right) \quad (40)$$

$$\sum_{u \in \mathcal{N}_i^-} \frac{(\mathbf{z}_u^{(l)})^T \mathbf{z}_i^{(l)} \mathbf{z}_u^{(l)}}{m - d_u}$$

$$= \left( \sum_{u \in \mathcal{U}} \frac{(\mathbf{z}_u^{(l)})^T \mathbf{z}_i^{(l)} \mathbf{z}_u^{(l)}}{m - d_u} \right) - \left( \sum_{u \in \mathcal{N}_i^+} \frac{(\mathbf{z}_u^{(l)})^T \mathbf{z}_i^{(l)} \mathbf{z}_u^{(l)}}{m - d_u} \right) \quad (41)$$

$$= (\mathbf{z}_i^{(l)})^T \left( \sum_{u \in \mathcal{U}} \frac{\mathbf{z}_u^{(l)} \mathbf{z}_u^{(l)}}{m - d_u} \right) - \left( \sum_{u \in \mathcal{N}_i^+} \frac{(\mathbf{z}_u^{(l)})^T \mathbf{z}_i^{(l)} \mathbf{z}_u^{(l)}}{m - d_u} \right) \quad (42)$$

The time and space complexity of calculating $\sum\limits_{i \in \mathcal{N}_u^+} (\mathbf{z}_u^{(l)})^T \mathbf{z}_i^{(l)} \mathbf{z}_i^{(l)}$ and $\sum\limits_{u \in \mathcal{N}_i^+} \frac{(\mathbf{z}_u^{(l)})^T \mathbf{z}_i^{(l)} \mathbf{z}_u^{(l)}}{m - d_u}$ in is $O(Ed)$, while the time and space complexity of calculating $(\mathbf{z}_u^{(l)})^T \left( \sum\limits_{i \in \mathcal{I}} \mathbf{z}_i^{(l)} \mathbf{z}_i^{(l)} \right)$ and $(\mathbf{z}_i^{(l)})^T \left( \sum\limits_{u \in \mathcal{U}} \frac{\mathbf{z}_u^{(l)} \mathbf{z}_u^{(l)}}{m - d_u} \right)$ is $O((n + m)d^2)$.

Therefore, the overall time and space complexity is $O((n+m)d^2 + Ed)$.

