# OpenReview forum: "Rankformer: A Graph Transformer for Recommendation based on Ranking Objective"
_ACM.org/TheWebConf/2025/Conference — WWW 2025 Poster_

### Official Review · Reviewer_p6re · 2024-11-20

**Novelty:** 3
**Technical Quality:** 3

**Review:**

The paper proposes Rankformer, a ranking-inspired recommendation model. The Rankformer layer is directly aligned with the ranking objective, which aims to elevate scores of the positive instances over the negative ones.


**Strengths**:
- S1: The proposed model directly optimizes the ranking-based objective, and leverages signals from negative instances during the aggregation process.
- S2: An empirical evaluation was conducted on four datasets against a wide variety of baseline methods, and the results highlight the advantage of Rankformer against current state-of-the-art methods.
- S3: A detailed analysis of Rankformer is provided in the paper, and Rankformer seems to be a rather efficient model as compared to LightGCN and so on.


**Weaknesses**:
- W1: The proposed model seems to be modifying existing graph-based methods, e.g., LightGCN, in a rather trivial manner, i.e. by adding various 'weights' to adjust the process of aggregating neighbouring nodes. For instance, as shown in Table 4, removing the 'benchmark term' does not cause any significant decrease to the model's performance. Similarly, as shown in Figure 5, adjusting the offset $\alpha$ affects the performance very insignificantly.
- W2: The motivation is somewhat unclear. In Figure 1, the authors try to demonstrate that "performance gains from stacking multiple layers of GNNs are limited". However, the comparison between LightGCN and Rankformer is done for "randomly initialized representations *without training*". This seems rather puzzling and does not say much about 'limited performance gains'. Furthermore, does this still hold after the models have been trained?

**Questions:**

Please clarify the motivation behind the proposed framework. Figure 1 seems to be a rather weak motivation (see W2).

**Reviewer Confidence:**

4: The reviewer is certain that the evaluation is correct and very familiar with the relevant literature

**Scope:**

4: The work is relevant to the Web and to the track, and is of broad interest to the community

---

### Official Review · Reviewer_rvb3 · 2024-11-29

**Novelty:** 6
**Technical Quality:** 5

**Review:**

**Summary**

This paper introduces Rankformer, a ranking-inspired transformer model designed explicitly for personalized recommendation systems. It addresses the gap between traditional model architectures and the inherently personalized ranking objective of recommender systems. Rankformer integrates a novel gradient-descent-inspired architecture with efficient computational strategies, enabling superior ranking performance while significantly reducing computational complexity. The proposed model demonstrates strong empirical results across multiple real-world datasets.

---

**Strengths**

1. **Alignment with Ranking Objective**: Rankformer’s architecture is directly inspired by the gradient of the ranking objective, ensuring its design aligns closely with personalized ranking requirements.
2. **Efficiency**: The proposed acceleration algorithm reduces the computational complexity of Rankformer, making it feasible for large-scale datasets without compromising performance.
3. **Comprehensive Evaluation**: Experiments on four diverse datasets demonstrate that Rankformer consistently outperforms sufficient state-of-the-art baselines.


---

**Weaknesses**

1. I find Figure 1 impressive, showing that Rankformer can achieve notable performance on the Ali-Display dataset even without training. In Table 1, the primary distinction between the attention mechanism proposed in Rankformer and the standard Transformer lies in the benchmark $b$ and the offset $\alpha$. However, in Table 4, the ablation study reveals that the benchmark $b$ has little impact on performance across the four datasets. Could the authors provide further explanation on what exactly makes Rankformer effective? If the benchmark $b$ contributes minimally, what specific aspects of the design or methodology drive its success?

2. Could you deeply explain why graph transformer baselines such as Nodeformer, DIFFormer, and SGFormer perform so poorly in this setting? Their performance seems to be significantly worse compared to their success in node classification tasks. Could this be because these models lack normalization coefficients, as highlighted in your ablation study?

3. From Equation 5, I understand that Rankformer might treat all items not clicked by a user as negative items. Is my understanding correct? If so, would designing a more reasonable selection strategy for negative items potentially improve the model's performance? This could address potential noise or redundancy in treating all unclicked items as negatives.

4. The authors describe Rankformer as a ranking-inspired transformer model. However, the purpose of rank loss is to ensure that positive items rank higher than negative items. In Equation 5, the positive and negative terms are still summed together. Could the authors further elaborate on the specific connection to the ranking-inspired concept? Is it only that more negative samples are incorporated during graph aggregation, or are there additional aspects that justify the ranking inspiration?


I am open to revise my score according to the rebuttal of authors.

**Questions:**

See weaknesses

**Reviewer Confidence:**

4: The reviewer is certain that the evaluation is correct and very familiar with the relevant literature

**Scope:**

4: The work is relevant to the Web and to the track, and is of broad interest to the community

---

### Official Review · Reviewer_HU1v · 2024-11-29

**Novelty:** 5
**Technical Quality:** 5

**Review:**

This paper proposes a novel attention-based model for recommender systems, which incorporates the idea of gradient descent of ranking objectives into the design of the model architecture. The proposed method has been shown to outperform existing methods based on GNNs and and graph transformers.

Pros:

1. The presentation of the paper is generally clear. The objective and the method to achieve that objective are clearly stated.

2. The exhaustive proof in Appendix theoretically supports the validity of the proposed method.

Cons:

1. The rationale behind the effectiveness of this method is missing. In Comparison win direct optimization in Section 3.2, the authors describe a very interesting statement that simulating multiple iterations of gradient descent leads to a more advantageous optimization in the advanced perspective of the optimization landscape. However, this claim needs theoretical analysis, or at least empirical observation on how this landscape looks like and how this optimization directions differ from iterating gradient descent multiple times in the existing methods like LightGCN.

2. There are grammar mistakes as follows:

- Please fix a grammar mistake in line 83: despite their increasingly sophisticated,

- grammar mistake in line 1091: reorganization -> reorganizing

**Questions:**

Does omitting the projection matrices for query, key, and value result in the better performance? Why do the authors omit these parameters in the Transformer model? For effectiveness or efficiency?

The other limitations are described above.

**Reviewer Confidence:**

3: The reviewer is confident but not certain that the evaluation is correct

**Scope:**

3: The work is somewhat relevant to the Web and to the track, and is of narrow interest to a sub-community

---

### Official Review · Reviewer_feRr · 2024-12-02

**Novelty:** 5
**Technical Quality:** 4

**Review:**

This paper proposes a framework called Rankformer, which is inspired by the gradient of ranking objectives. This means that the model's architecture directly mimics the gradient descent steps in the ranking optimization process.

**Pros:**
1. Designing the model architecture based on insights from ranking objectives is novel and effective.
2. This paper is well-written, and the motivation is easy to understand.
3. The conducted experiments seem to demonstrate the effectiveness of the proposed method.

**Cons:**

See questions.

**Questions:**

1. In Eq. (1), what do $d_u$ and $m$ represent? It seems that the authors did not clearly mention the meaning of these two variables.

2. Regarding the fast implementation, from Eq. (8) to Eq. (10), both equations require calculating the sum of positive/negative item embeddings. How is the computational complexity reduced?

3. Since the forward processing of Rankformer indeed considers the ranking process, is training necessary? Have the authors tried stacking a larger number of Rankformer layers to see if it can achieve the performance of a trained Rankformer with a lower number of layers?

**Reviewer Confidence:**

3: The reviewer is confident but not certain that the evaluation is correct

**Scope:**

3: The work is somewhat relevant to the Web and to the track, and is of narrow interest to a sub-community

---

### Official Review · Reviewer_ZUfg · 2024-12-03

**Novelty:** 5
**Technical Quality:** 4

**Review:**

The paper introduces Rankformer, a novel Transformer-based recommendation model that fundamentally integrates a ranking objective into its architecture. Rankformer innovatively combines global user-item interactions with benchmark and offset terms to explicitly align representation learning with ranking performance. This unique design ensures that the model evolves embeddings in a direction directly guided by the ranking objective.

Strengths:
1.	Rankformer directly incorporates ranking principles into its architecture by simulating the gradient descent process of ranking loss, ensuring alignment with personalized ranking goals.
2.	The paper proposes an acceleration algorithm that significantly reduces the computational complexity of the model. By reducing complexity from quadratic to linear with respect to positive instances, Rankformer becomes scalable and practical for large-scale datasets.
3.	Experimental results demonstrate that Rankformer consistently outperforms leading baseline models across multiple datasets in terms of ranking metrics like NDCG@20 and Recall@20.


Weaknesses:
1.	The model emphasizes the role of negative instances, but the article lacks a detailed explanation of the negative sampling strategy. Are negative samples selected randomly, or are they based on some heuristic?
2.	The article briefly mentions contrastive learning but does not fully explore its potential integration with Rankformer. Could incorporating contrastive objectives further enhance its performance?
3.	The benchmark and offset terms are presented as innovative components that help distinguish positive and negative instances and align the model with the ranking objective. Although ablation studies validate their importance, the reasons for unexpected performance changes on some datasets (e.g., Amazon-Kindle) are not thoroughly explained. For instance, why does removing the benchmark term sometimes improve performance?
4.	The manuscript introduces Rankformer, which shares similarities with the model presented in “RankFormer: Listwise Learning-to-Rank Using Listwide Labels”. Both utilize Transformer architectures to enhance ranking performance, but their approaches differ in incorporating ranking objectives. Please provide a comparison to highlight how the proposed method contrasts with the listwise label-based optimization approach.
5.	Sharing the code for Rankformer would greatly support reproducibility and allow researchers to validate results, test the model on diverse datasets, and extend the methodology in innovative ways.

**Questions:**

Please check weakness 1-4.

**Reviewer Confidence:**

4: The reviewer is certain that the evaluation is correct and very familiar with the relevant literature

**Scope:**

4: The work is relevant to the Web and to the track, and is of broad interest to the community